# Oncogenic Impact of TONSL, a Homologous Recombination Repair Protein at the Replication Fork, in Cancer Stem Cells

**DOI:** 10.3390/ijms24119530

**Published:** 2023-05-31

**Authors:** Hani Lee, Sojung Ha, SeokGyeong Choi, Soomin Do, Sukjoon Yoon, Yong Kee Kim, Woo-Young Kim

**Affiliations:** 1College of Pharmacy, Sookmyung Women’s University, Seoul 04310, Republic of Korea; 2Muscle Physiome Research Center, Sookmyung Women’s University, Seoul 04310, Republic of Korea; 3Department of Biological Sciences, Sookmyung Women’s University, Seoul 04310, Republic of Korea; 4Research Institute of Pharmacal Research, Sookmyung Women’s University, Seoul 04310, Republic of Korea

**Keywords:** TONSL, CSC, MYC, double-strand DNA damage repair, homologous recombination repairs

## Abstract

We investigated the role of TONSL, a mediator of homologous recombination repair (HRR), in stalled replication fork double-strand breaks (DSBs) in cancer. Publicly available clinical data (tumors from the ovary, breast, stomach and lung) were analyzed through KM Plotter, cBioPortal and Qomics. Cancer stem cell (CSC)-enriched cultures and bulk/general mixed cell cultures (BCCs) with RNAi were employed to determine the effect of *TONSL* loss in cancer cell lines from the ovary, breast, stomach, lung, colon and brain. Limited dilution assays and ALDH assays were used to quantify the loss of CSCs. Western blotting and cell-based homologous recombination assays were used to identify DNA damage derived from TONSL loss. *TONSL* was expressed at higher levels in cancer tissues than in normal tissues, and higher expression was an unfavorable prognostic marker for lung, stomach, breast and ovarian cancers. Higher expression of *TONSL* is partly associated with the coamplification of *TONSL* and *MYC*, suggesting its oncogenic role. The suppression of *TONSL* using RNAi revealed that it is required in the survival of CSCs in cancer cells, while BCCs could frequently survive without *TONSL*. *TONSL* dependency occurs through accumulated DNA damage-induced senescence and apoptosis in *TONSL*-suppressed CSCs. The expression of several other major mediators of HRR was also associated with worse prognosis, whereas the expression of error-prone nonhomologous end joining molecules was associated with better survival in lung adenocarcinoma. Collectively, these results suggest that TONSL-mediated HRR at the replication fork is critical for CSC survival; targeting TONSL may lead to the effective eradication of CSCs.

## 1. Introduction

Cancer stem cells (CSCs) are a subset of tumor cells that mediate the initiation, resistance and metastasis of tumors [1]. Although the origin and definition of CSCs is still not clear, the undifferentiated stem cell-like characteristics of CSCs may provide an advantage in the acquisition of new mutations to survive and migrate/adapt to another tissue environment [2]. Previously, we screened novel therapeutic targets that eliminated CSCs efficiently in a glioblastoma multiform (GBM) cell line and found that several lipid metabolism enzymes are essential for CSC survival but are not essential in bulk cultured cells (BCCs) [3]. In the same study, TONSL (NFκBIL2, Tonsoku-like protein) was found to be a candidate selective target for CSC elimination.

TONSL forms a complex with MMS22L (MMS22-like) that plays an important role in maintaining genome integrity [4]. The complex mediates homologous recombination repair of double-strand breaks (DSBs) at stalled or collapsed replication forks [5]. At the damaged site of a replication fork, TONSL–MMS22L loads Rad51 to replace RPA [6] with the help of histone chaperones [7] and the epigenetic modification of histones [8]. Biallelic mutation of *TONSL* causes spondyloepimetaphyseal dysplasia (SEMDSP) [9,10], probably due to the delay of replication, which is associated with several malignancies [11].

Genomic integrity is maintained by many DNA damage repair (DDR) molecules. As cancer is a disease caused by accumulated DNA damage, loss of the DDR function is an important determinant of carcinogenesis [12]. Therefore, both germline and somatic mutational defects in DDR genes can act as strong drivers of carcinogenesis [13]. In particular, two major homologous recombination repair (HRR) genes, *BRCA1* and *BRCA2*, become nonfunctional in cancer through loss of function (LOF) mutations or hypermutations in the promoter region [14]. This suggests that LOF mutations of other HRR genes may contribute to tumorigenesis as well. Indeed, other core HRR-associated genes (*BARD1*, *PALB2*, *FANCC*, *RAD51C* and *RAD51D*) are frequently lost in many cancers [15]. Interestingly, it has been noted that the *TONSL* gene is amplified in several cancers [16], suggesting an oncogenic role different from those of other well-studied tumor-suppressive HRR genes in cancer or carcinogenesis.

In this study, we investigated whether the *TONSL* gene plays an important role in cancer and CSCs using RNAi and CSC enrichment cultures in addition to bioinformatics analyses of various public databases. We demonstrate that TONSL is enriched in cancer tissues versus normal tissues and that higher expression is associated with worse prognosis. The previous finding that *TONSL* may be essential in CSCs was demonstrated again in several cancers. The dependency of *TONSL* and other HRR genes suggests that HRR in the replication fork may be important in CSCs and could be a target for CSC therapy.

## 2. Results

Because we previously noted that GBM CSCs are more vulnerable to the loss of *TONSL* than bulk cultured cells (BCCs), we investigated whether *TONSL* plays an important role in cancer prognosis in other cancers (Figure 1). The association of *TONSL* mRNA expression with the survival of cancer patients was analyzed via the KM plotter online tool [17]. Higher expression of *TONSL* mRNA was significantly associated with worse overall survival (OS) in lung adenocarcinoma, gastric cancer (intestinal and diffuse), breast cancer (luminal A, luminal B and HER2-positive) and ovarian cancer (endometrial and serous) (hazard ratios were greater than 1.2). Lung squamous cell carcinoma (SQCC) showed a similar pattern, but without statistical significance. Interestingly, the Hazardous Ratio was less than 0.5 for the basal type of breast cancer, suggesting that higher expression of *TONSL* mRNA is associated with better prognosis. These data showed that *TONSL* expression may be a prognostic marker for worse survival in some subsets of tumors from the lung, breast, ovary and stomach, suggesting an oncogenic role in them. We also examined recurrence-free survival (RFS) in the same manner, and the pattern of RFS according to the expression of TONSL mRNA was similar to that of OS, except for luminal B breast cancer, which had the opposite result (Appendix A).

Because this finding suggests pro-cancer roles of *TONSL* expression, we questioned whether the expression of *TONSL* increases in cancer. We extracted mRNA expression data in cancers and corresponding normal tissues from the Pan-Cancer Atlas Study [18] in The Cancer Genome Atlas (TCGA) and analyzed them using Qomics (https://qomics.sookmyung.ac.kr/, accessed on 1 December 2022) [19] or cBioportal (https://www.cbioportal.org/, accessed on 1 December 2022) [20,21]. The expression of *TONSL* was significantly higher in all tumors examined: lung, stomach and breast (Figure 2A and Appendix A). Because no normal tissue data were available for ovarian tumors, we could not analyze the relative expression between tumors and normal tissues for this cancer. Several oncogenes were found to be amplified, leading to an increase in mRNA expression. Therefore, we assessed the copy number variation (CNV) of *TONSL* in the cancers indicated in Figure 1. Interestingly, more than 5% of the tumors from all of the examined cancer tissues showed greater *TONSL* gene amplification (Appendix A). Among those, more than 25% of ovarian cancers harbored gene amplification of *TONSL*. As TONSL requires MMS22L to mediate repair in the replication fork, we also examined gene amplification of *MMS22L*. *MMS22L* gene amplification was very rare in all cancers assessed, and point mutations were the most common variations in stomach and lung adenocarcinomas. We then examined whether increased expression of *TONSL/MMS22L* mRNA is associated with *TONSL* and *MMS22L* CNV (Figure 2B). In ovarian cancers, 1/3 of the *TONSL*-amplified tissues expressed higher *TONSL* mRNA (of 32% of tumors with *TONSL* gene amplification, only 12% highly expressed *TONSL* mRNA). Approximately 8% of total tumors without gene amplification expressed TONSL mRNA at high levels. In other organ tumors (stomach, breast and lung adenocarcinoma and squamouscarcinoma), the majority of tumors that highly expressed *TONSL* mRNA did not exhibit *TONSL* gene amplification (14.7%, 12.5%, 18.3% and 14.0% respectively), and only a small portion of those tumors were accompanied with *TONSL* gene amplification (2.2%, 5.0%, 1.8% and 1.8%, respectively). We also evaluated *MMS22L* mRNA expression, point mutations and CNV in the same tumors. Only 4–8% of tumors had higher *MMS22L* expression without CNV or mutation. This is somewhat surprising because TONSL functions as a complex with MMS22L. This suggests that *TONSL* amplification may not be directly caused by cooperative function with *MMS22L*, or that the amplification itself is not an oncogenic driver.

Accordingly, we sought to determine why amplification of the *TONSL* gene frequently occurs in ovarian cancer, and its impact. One of the main mechanisms by which proto-oncogenes become oncogenes for carcinogenesis is amplification. The most frequently amplified genes in all cancers are *MYC*, *EGFR* and *ERBB2* (HER2). We explored the portion of cancers carrying these gene amplifications in the same dataset in TCGA. TCGA Pan-Cancer Atlas Study [18] data were analyzed in cBioportal. The *MYC* gene was amplified in 9% of total tumors and most amplified (32%) in ovarian cancer (serous). *ERBB2* amplification was found in 6% of total tumors and most amplified in gastric and breast cancers (11% and 10%, respectively). *EGFR* amplification was detected in 7% of total tumors; glioblastoma presented an amplification frequency that was greater than 40%. *TONSL* is located at the end of the long arm of chromosome 8 (Ch 8q24.3), close to where the *MYC* gene is located (Ch 8q24.21). It has been reported that the long arm of chromosome 8 (Ch 8q) is the most highly amplified [22] in the cancer genome. As *MYC* is one of the most potent oncogenes, genes near *MYC* might be coamplified as passenger genes, and *TONSL* may be one of them. We then compared amplifications around *MYC*, *ERBB2* and *EGFR* in the whole chromosome view in corresponding highly amplified cancer tissues (Figure 3A). For *MYC* amplification, the whole region of Ch 8q was assessed as being largely amplified; the *TONSL* gene, 17 Mb from the *MYC* locus and near the telomere, was also amplified (the amplified area is shown in red throughout the arm in the figure). This may be caused by some positive or supportive effect derived from the coamplification of these genes with *MYC*. In contrast, *ERBB2* and *EGFR* amplifications were exclusively focal, suggesting that neighboring genes may not have a meaningful positive impact on the oncogenic roles of *EGFR* and *HER2*. However, it is still possible that coamplification at Ch 8q simply occurs because these regions are close in proximity to the potent oncogene *MYC*. We then explored the correlation between the distance from *MYC* and the coamplification rate. We collected CNV data for many genes on Ch 8q and around the MYC gene. Amplification of genes between 18 Mb up- and downstream of the *MYC* gene (Ch 8q24.21) in ovarian cancer was examined (Figure 3B, left). In total, 78% of *MYC* amplification cases also showed amplification of *TONSL*, which is approximately 18 Mb from *MYC* and near the telomere. Interestingly, amplification of *PKHD1L1* and other genes at a similar distance from *MYC* but in the opposite direction from *TONSL* was observed in only 60% of *MYC*-amplified cancers. Similar analysis was conducted with genes close to the *EGFR* in the glioblastoma (Figure 3B, right). The list of genes, the distance from *MYC* or *EGFR* and the coamplification rate are shown in Appendix A. These data support the hypothesis that amplification of genes between *MYC* and the telomere (including *TONSL*, right side of *MYC* in the chromosome shown in Figure 3B) may play a supportive and/or additive role in *MYC*-driven carcinogenesis. Several genes in this region mediate replication and repair, which may contribute to the oncogenic role of *MYC*.

Based on the results, we hypothesized that the higher *TONSL* expression in tumors compared to normal tissues, and the associated poor prognosis, is not caused by gene coamplification with *MYC* alone. However, we sought to determine the effect of highly amplified *TONSL* in ovarian cancer. Because ovarian cancer showed the highest gene amplification frequency of *TONSL*, and the gene expression was associated with poor prognosis, we examined whether loss of *TONSL* changes cancer cell behavior using ovarian serous adenocarcinoma cell lines (Figure 4A) transduced by sh*TONSL*- or shNC-expressing lentivirus. Based on the results of monolayer bulk cultured cells (BCCs) and CSC-enriched sphere cultures, *TONSL* expression is required for growth in both cell populations. Notably, when *TONSL* was suppressed, secondary CSC sphere formation was completely disrupted in the OVCAR8 cell line. This suggests a critical role in CSC maintenance or survival. We also confirmed the requirement of *TONSL* in CSCs derived from lung (H1299 and H460) and breast (MDAMB468 and MDAMB231) cancer cell lines, of which we observed the clinical importance of in Figure 1. Interestingly, TONSL was dispensable for BCC survival; the requirement of TONSL was always stricter in CSC than that in BCC. We also found that TONSL is required for CSCs in glioblastoma (U87MG and LN229) and colon cancer (HCT15 and HT29). (Figure 4B). Representative images of CSCs are shown in Figure 4C.

We also confirmed that the lost cells in CSC spheres due to TONSL depletion included the CSC population, using the limited dilution assay [23] and a specific marker of CSCs, ALDH1 (Figure 5A,B) [23]. SCD1, which is essential for CSC survival, was used as the positive control [24,25]. We next investigated the mechanism by which CSCs are eliminated when TONSL is absent. Interestingly, TONSL-depleted OVCAR8 CSCs did not exhibit clear apoptosis in FACS analysis, but that of BCCs did. However, senescence increased dramatically in TONSL-depleted CSCs (Figure 5C)**,** suggesting that senescence is the main mechanism eliminating the CSC population in these ovarian cancer cells and can be different from the mechanism of BCC loss. We then tested whether the importance of TONSL in CSCs is related to its well-known role in the replication fork, which is mediated by its partner MMS22L (Figure 6). siRNAs targeting MMS22L suppressed growth of HCT15 CSCs but had no effect on that of monolayer-cultured BCCs. This result was the same as the results acquired with *TONSL* knockdown in Figure 4 with HCT15. We also confirmed that CSCs in glioblastomas required MMS22L. These results strongly suggest that the complex composed of TONSL and MMS22L, which mediates HRR in stalled replication forks, plays the same pivotal role in CSC maintenance in colon cancer cells. Surprisingly, BCCs may circumvent the loss of these factors in some cancer cell lines. We explored the mechanism that resulted in differential responses in BCCs and CSCs. In the colon cancer cell HT29 (Figure 7A), the siRNA targeting *TONSL* increased the expression of G2/M cyclins (Cyclin B and Cyclin A) in both cell groups (BCC and CSC). γH2AX, the surrogate marker for dsDNA breaks, was also increased in CSCs with *TONSL* knockdown, whereas the downstream signaling molecules p-chk1 and p-chk2 were increased in both BCCs and CSCs. We also detected PARP-1 cleavage in CSCs, which was not detectable in BCCs. After the loss of TONSL, more apoptotic cells were detected among CSCs by flow cytometry analysis. We also confirmed that γH2AX was increased by the suppression of *TONSL* in ovarian cancers (Figure 7B). We generated an HRR GFP reporter system in U87MG using pHPRT-DRGFP [26]. The reporter-HRR efficiency was diminished by knockdown of *TONSL* in both BCCs and CSCs of the U87MG cell line, suggesting that the HRR mechanism itself was damaged in both BCCs and CSCs (Figure 7C).

We hypothesized that TONSL loss-driven accumulation of dsDNA breaks is solved in the BCC population somehow, through error-prone nonhomologous end joining (NHEJ) or microhomology-mediated end joining (MMEJ), but accumulates in CSCs, preventing their survival. If this is true, more dsDNA breaks caused by treatment with DNA-damaging anticancer drugs should render CSCs more dependent on the TONSL/MMS22L complex. We tested this hypothesis with camptothecin (Cpt), which induces dsDNA breaks by binding to topoisomerase I [27]. Treatment of U87MG cells with Cpt showed that Cpt suppressed CSC sphere growth (Figure 8). The Cpt cytotoxicity (at 2 nM and 10 nM) was further enhanced by *TONSL* knockdown in CSCs. However, the cytotoxic effect of Cpt on BCCs was not increased by *TONSL* knockdown at the same concentration or at a much higher concentration. Therefore, it seems that TONSL plays a more critical role in genome integrity maintenance for survival in CSCs than in BCCs (Figure 8).

The MMS22L–TONSL complex resolves the stalled replication fork through homologous recombination repair (HRR). Many HRR mediators, such as BRCA1, BRCA2, BLM and Rad51, are directly involved in the replication fork resolution process. Accordingly, we determined whether the expression of these factors is also associated with poor prognosis of cancer, as observed with *TONSL* in Figure 1. Interestingly, the expression of *BRCA1*, *BRCA2*, *BLM* and *RAD51* was significantly associated with poor prognosis in lung adenocarcinoma (Figure 9). Strikingly, the factors that exclusively mediate NHEJ, *53BP1*, *LIG4*, *ARTEMIS* and *XRCC4* were inversely associated with poor prognosis. Overall, higher expression in tumor tissue was associated with longer survival in lung adenocarcinomas, which was the opposite effect to that of *TONSL*.

## 3. Discussion

In this study, we identified that the gene of a replication fork HRR mediator, TONSL, is amplified and transcriptionally upregulated in several cancers. Higher expression was associated with a worse survival rate in several cancer types. In addition, the loss of TONSL resulted in selective depletion of CSCs in the tested cell lines, including colon cancer, ovarian cancer and glioblastoma lines. CSC loss occurs through senescence and/or apoptosis, probably derived from accumulated DNA damage in the CSC population; in contrast, BCCs can manage the situation and survive in a subset of cell lines. The selective toxicity of TONSL loss to CSCs versus BCCs was more evident when dsDNA breakage was induced by anticancer therapeutic agents. As with TONSL, several HRR mediators at the fork are also associated with poor prognosis in lung adenocarcinoma. Conversely, NHEJ mediators are mostly associated with better survival. Because normal stem cells may invoke error-free HRR over error-prone NHEJ, CSCs may still maintain the characteristics of normal stem cells [28].

CNVs are one of the most important classes of genomic mutations related to carcinogenesis [29]. CNVs involve deletions or amplifications of large contiguous segments of the genome, including tumor-suppressive genes and oncogenic genes, respectively [29]. Amplification of genes is occasionally associated with transcriptional upregulation of the amplified gene [30]. Chromosome 8q is one of the most amplified segments in cancer [31], and the gene *TONSL*, which is located on the same arm, is also amplified in many cancers (Appendix A). However, as demonstrated in Figure 2 and Figure 3, enhanced transcription of *TONSL* was detected in cancers regardless of *TONSL* amplification.

Unrepaired lesions of DNA base adducts, mismatched bases and single-strand breaks can generate dsDNA breaks (DSBs) at the DNA replication fork. Unrepaired DSBs cause multiple chromosome instability (CIN) [32], which is associated with poor prognosis, metastasis and therapeutic resistance. When DSB lesions occur, HRR, which preserves genomic integrity without errors, is invoked. However, when HRR is not available, cells activate other pathways despite junction site errors. The HRR gene group is one of the most frequently defective genes in cancer [33]. Loss-of-function (LOF) germline mutations in HRR genes, such as *BRCA1* and *BRCA2*, significantly contribute to the elevated risk of cancer in heterozygous carriers [14,34]. Sporadic mutations of these genes and hypermethylation of their promoters, leading to lowered expression, also contribute significantly to carcinogenesis. In addition to *BRCA1* and *BRCA2*, many other HRR mediators (*ATM*, *BRIP1*, *CHEK2*, *NBS1* or *RAD51C*) are frequently defective in cancer [14,34]. In other words, loss of the DDR in cancer cells may contribute to the efficacy of DNA-damaging therapeutics [35]. Specifically, homologous recombination repair (HRR) gene defects contribute to the efficacy of some DNA-damaging reagents, such as cisplatin and PARP inhibitors [36,37]. Therefore, HRR gene loss can contribute to both oncogenic processes and therapeutic biomarkers. Conversely, elevated expression of the essential HRR gene *TONSL* in cancer versus normal tissues and its association with poor prognosis in many major cancer types is distinguished from the characteristics of many other HRR gene alterations.

*TONSL* is located on chromosome 8q24.3, in the region most often amplified in human cancers (up to 40% in breast cancer) [38] and adjacent to the strong oncogenic driver *MYC* (c-myc), which is located at 8q24.2. Recurrent amplification of this large region of chromosome 8q suggests that multiple genes in this segment, in addition to *MYC*, support the “driver” role in oncogenesis [19]. Figure 3 also shows that many genes (spanning over 30 Mb around *MYC*) were coamplified with *MYC*. Interestingly, the genes closer to telomeres (including *TONSL*) than to centromeres were more frequently coamplified with *MYC* (80% vs. 60%, respectively) (Figure 3). Although we do not have sufficient data to support the idea that *TONSL* itself is indeed a “driver” in carcinogenesis, it is possible that multiple genes in these regions together contribute to *MYC*-driven oncogenesis. Several reported pro-proliferation genes, such as the potassium channel gene *KCNK* [39], the DNA helicase *RECQL4* [40] and the chromatin modulator *PARP10*, are also located at 8q24.3 and become coamplified. Collectively, it may be a reasonable hypothesis that some of these genes cooperate with *MYC* to enhance its oncogenic “driver” effect.

Based on this hypothesis, we questioned the impact of *TONSL* loss on ovarian cancer, lung cancer, breast cancer, colon cancer and glioblastoma. *TONSL* loss resulted in depletion of the CSC population in many cell lines, whereas this loss had a less severe effect in the BCC population of several cell lines. In those several cell lines, CSCs required *TONSL* for survival, showing that *TONSL* is not dispensable for CSC survival. As TONSL-mediated HRR in the replication fork is essential, to preserve genome integrity in the repair process, the requirement of *TONSL* should be the same in all cancer cells. Nevertheless, the necessity of HRR over other repair mechanisms may be more important in normal stem cells, and CSCs may be reminiscent of these stem cell characteristics. However, when HRR is not available, non-CSC cancer cells may survive through error-prone non-HRR repair processes. These processes at the stalled replication fork include MMEJ, which is mediated by DNA polymerase theta [41].

The preference for HRR in normal germ/stem cells has been suggested multiple times. For example, stem cells barely express NHEJ genes but they do express HRR genes. Stem cells have a longer S phase to help the HRR process [42]. In addition, the CSC preference for HRR has also been suggested in breast and gastric CSCs [43,44]. We also recently suggested that HRR may be indispensable for colon CSC survival [45]. The current study shows that TONSL is indispensable to many tissue-originated CSCs, which further supports the idea that HRR can be a weak point for CSC survival.

In this study, consistently, the replication fork HRR mediator TONSL was shown to be essential for CSC survival. The elimination of CSCs is clinically important, and HRR may be a therapeutic target for this approach. Many HRR-targeting drugs that are being developed may be tested in this regard. However, the discrepancy in mutation patterns between many HRR genes and *TONSL* suggests that the situation is more complicated. In addition, the *MMS22L* mutation pattern, which is rarely amplified in cancer, should be considered when evaluating the significance of the oncogenic role of *TONSL*. It is also necessary to consider the possibility that MMS22L plays an independent role from TONSL [46].

In summary, we demonstrated the oncogenic role of the replication fork HRR gene *TONSL*, which is frequently amplified along with *MYC*, in the maintenance of CSC. Our findings suggest that this gene is a potential target in CSC elimination therapy. The results also indicate that CSC-preferred HRR is a vulnerable target in CSC, which is reminiscent of the normal stem cell characteristics that protect the genome integrity of stem cells. Further investigation regarding the clinical use of these potential target processes is warranted.

While this manuscript was being revised for publication, a paper was published demonstrating that TONSL is an immortalizing oncogene in breast cancer oncogenesis [47]. The results of the present study will further increase the understanding of the role of TONSL in cancer stem cells specifically and provide a rationale for targeting TONSL to treat cancer.

## 4. Materials and Methods

### 4.1. Cell Lines

The HEK293T cell line was obtained from the Korean Cell Bank (Seoul, Republic of Korea). The human ovarian cancer cell lines IGR-OV1 and OVCAR8, the human breast cancer cell lines MDAMB231 and MDAMB468, the human lung cancer cell lines H1299 and H460, and the human colon cancer cell lines HT29 and HCT15 were obtained from the National Cancer Institute. The human glioblastoma cell lines U87MG and LN229 were obtained from ATCC (Manassas, VA, USA). All cells were cultured in RPMI-1640 or DMEM (high glucose) medium (Thermo Fisher Scientific, Waltham, MA, USA) supplemented with 10% fetal bovine serum and 1% penicillin-streptomycin (all from Thermo) at 37 °C in a humid atmosphere containing 5% CO_2_. To enrich the cancer stem cells, cells were cultured in suspension in DMEM/F12 medium supplemented with B27 supplement (Thermo), 20 ng/mL EGF and 40 ng/mL FGF (all from Thermo) on poly-HEMA-coated culture dishes (Millipore-Sigma, Burlington, MA, USA), as previously described [39].

### 4.2. RNAi

siRNAs for *TONSL* and *MMS22L* were purchased from IDT (Coralville, IA, USA) and Genolution (Seoul, Republic of Korea), respectively. The sequences of each siRNA are listed in Appendix A. siRNA transfections were performed using Lipofectamine RNAiMAX (Thermo Fisher Scientific, Waltham, MA, USA) according to the manufacturer’s protocol. Cells were assayed 3–5 days after transfection. *TONSL*-specific shRNAs in the lentiviral pLKO.1 vector (TRCN0000424634, #1; TRCN0000424077, #2; and TRCN0000424443, #3) were obtained from Sigma-Aldrich (St. Louis, MO, USA), and the pLKO.1 vector was used as a control. Lentiviruses were produced in HEK293T cells by co-transfecting the shRNA-expressing vector, pMD2.G (Addgene, Watertown, MA, USA, #12259), and psPAX2 (Addgene) using jetPRIME (Polyplus-transfection, Illkirch, France) according to the manufacturer’s protocol. Cells were transduced with 5 μg/mL hexadimethrine bromide (Sigma). Two days later, the cell extract or RNA was used for western blotting or quantitative reverse transcription PCR to confirm the knockdown effect (Appendix A).

### 4.3. Western Blotting

Cells were lysed with RIPA buffer supplemented with a protease inhibitor cocktail (Sigma-Aldrich, Burlington, MA, USA), and protein concentrations in the extracts were measured using the BCA assay (Thermo Fisher Scientific, Waltham, MA, USA). Equal amounts of proteins were separated by SDS–polyacrylamide gels and transferred onto PVDF membranes (Millipore-Sigma). The membrane was blocked with 5% non-fat dry milk in Tris-buffered saline containing 0.04% Tween-20 (TBST) and incubated with primary antibodies overnight at 4 °C. Antibodies against Cyclin A (#4656), Cyclin B (#4138), γH2AX (#9718), p-Chk1 (#2348), and p-Chk2 (#2197) were purchased from Cell Signaling Technology (Danvers, MA, USA), while PARP (sc-7150), p53 (SC-126), GAPDH (SC-32233), β-actin (SC-130657) and α-Tubulin (SC-23948) were obtained from Santa Cruz Biotechnology (Dallas, TX, USA). The antibody against TONSL (ab101898) was purchased from Abcam (Cambridge, UK). The membranes were then washed with TBST and incubated with horseradish peroxidase-conjugated secondary antibodies (Jackson Immuno Research, Philadelphia, PA, USA) for 1 h at room temperature. After washing with TBST, the immunoreactive bands were detected using ECL and visualized on X-ray films or with an Image 680 LAS (GE healthcare, Amersham, UK) imaging system.

### 4.4. Apoptosis Detection

For apoptosis detection, cells were transduced with shRNA, and after 72 h they were trypsinized and stained with FITC Annexin V and propidium iodide (PI) using the FITC Annexin V apoptosis detection kit from BD Biosciences (Franklin Lakes, NJ, USA). The stained cells were then analyzed using a FACSCalibur flow cytometer from BD Bioscience.

### 4.5. Senescence Measurement

Senescent cells were measured using the CellEvent™ Senescence Green Flow Cytometry Assay Kit (Thermo). Briefly, after shRNA transduction for 72 h, the cells were treated with 100 nM bafilomycin A1 for 1 h at 37 °C. Subsequently, the cells were incubated with C12FDG to a final concentration of 33 μM for 1 h. After harvesting, the cells were analyzed for senescence-associated β-galactosidase activity, which was indicated by raised green fluorescence, using a FACSCalibur flow cytometer.

### 4.6. MTS Assay for BCC and Sphere Counting for CSC

The cell viability of BCC was assessed using the CellTiter 96 AQueous One Solution Cell Proliferation Assay kit (Promega, Madison, WI, USA). SiRNA-transfected cells were seeded at a density of 5 × 10^3^ cells/200 μL into each well of a 96-well plate and cultured for 4 days. For drug treatment, the cells were then exposed to the indicated concentrations of Camptothecin or vehicle control (DMSO, final concentration below 0.1%). After 4 days of treatment, 10 μL of the MTS reagent was added to each well, and the plate was incubated at 37 °C in a humid atmosphere containing 5% CO_2_ for 90 min. The absorbance was measured at 450 nm using a 96-well microplate reader. Both media were supplemented with 1% penicillin and streptomycin (Welgene, Gyeongsan, Republic of Korea). CSC was cultured in 96 wells coated with POLYHEMA at a density between 250/well and 2000/well depending on cell lines. After 5–7 days, when the control cells had made 20–100 spheres with diameters greater than 100 µmeter, the samples were fixed with formaldehyde and the spheres (>100 µm) were counted under a microscope.

### 4.7. Quantitative Reverse Transcription PCR (qRT-PCR)

The total RNA was extracted from cells using TRIsure (Bioline, London, UK), and cDNA synthesis was performed using 2 µg of total RNA and Superscript II reverse transcriptase (Thermo). The reaction conditions involved incubation at 45 °C for 10 min, followed by 95 °C for 10 min, and then 40 cycles of amplification at 95 °C for 15 s and 60 °C for 1 min. SYBR Green (SensiFAST SYBR Hi-ROX, Bioline, London, UK) was used to quantify the PCR product, and the StepOnePlusTM Real-Time PCR system (Applied Biosystems, Waltham, MA, USA) was employed for detection. The primer sequences used are listed in Appendix A. Relative mRNA expression levels were determined using the 2^−ΔΔCT^ method, with normalization to the expression levels of the housekeeping gene GAPDH.

### 4.8. Limiting Dilution Assay

The cells were adapted to si*TONSL* and plated in 96-well poly-HEMA-coated plates at various seeding densities (1–128 cells per well) containing sphere culture medium. After 7 days, the fraction of wells not containing spheres for each plating density was counted under a phase-contrast microscope. The data were then plotted against the number of cells per well.

### 4.9. ALDEFLUOR Assay

For the ALDEFLUOR assay, we utilized the Aldefluor kit (Stem cell Technologies, Vancouver, BC, Canada) to detect ALDH activity in cancer stem cell populations. Dissociated cells from CSC were incubated in assay buffer containing 1 μM of ALDH substrate for 30 min at 37 °C. As a negative control, a portion of ALDH substrate-treated cells were resuspended in buffer containing 15 μM of the specific ALDH enzyme inhibitor DEAB. ALDH-positive cells were quantified using FACS Calibur cytometry. The desired ALDH-positive cell population was determined based on the ALDH-positive regions set by DEAB-treated cells, which served as the control.

### 4.10. HRR Reporter Assay

The pHPRT-DRGFP reporter plasmid (#26476, AddGene, Cambridge, UK) was transfected into U87MG cells using Lipofectamine 2000. Single cells were seeded into 96-well plates, and Puromycin was added to establish stable clones at a final concentration of 1 μg/mL. The selected clone was then plated in a 6-well plate and transfected with 1 μg of I-SCEI plasmid (#26477, AddGene, Cambridge, UK) and 30 nM of si*TONSL*. After 72 h, cells were collected, and GFP-expressing cells were quantified using FACSCalibur cytometry.

### 4.11. Public Clinical Database Analyses

Publicly available clinical database analyses were conducted using data from The Cancer Genome Atlas (TCGA). mRNA expression in tumors and normal tissues was analyzed using Qomics (http://qomics.sookmyung.ac.kr/). Additionally, genome copy number variations and mRNA expression data from the TCGA Pan-cancer Atlas Study were analyzed using cBioportal. To examine the association between mRNA expression and patient survival, Kaplan–Meier survival curves were generated using KM Plotter (https://kmplot.com/analysis/). The “auto selection best cutoff” option was utilized to split patients into high- and low-expression groups based on the metadata of gene chips. The statistical significance was evaluated using the *p*-value provided by the corresponding analysis program portal. A *p*-value lower than 0.05 was considered statistically significant.

## Figures and Tables

**Figure 1 ijms-24-09530-f001:**
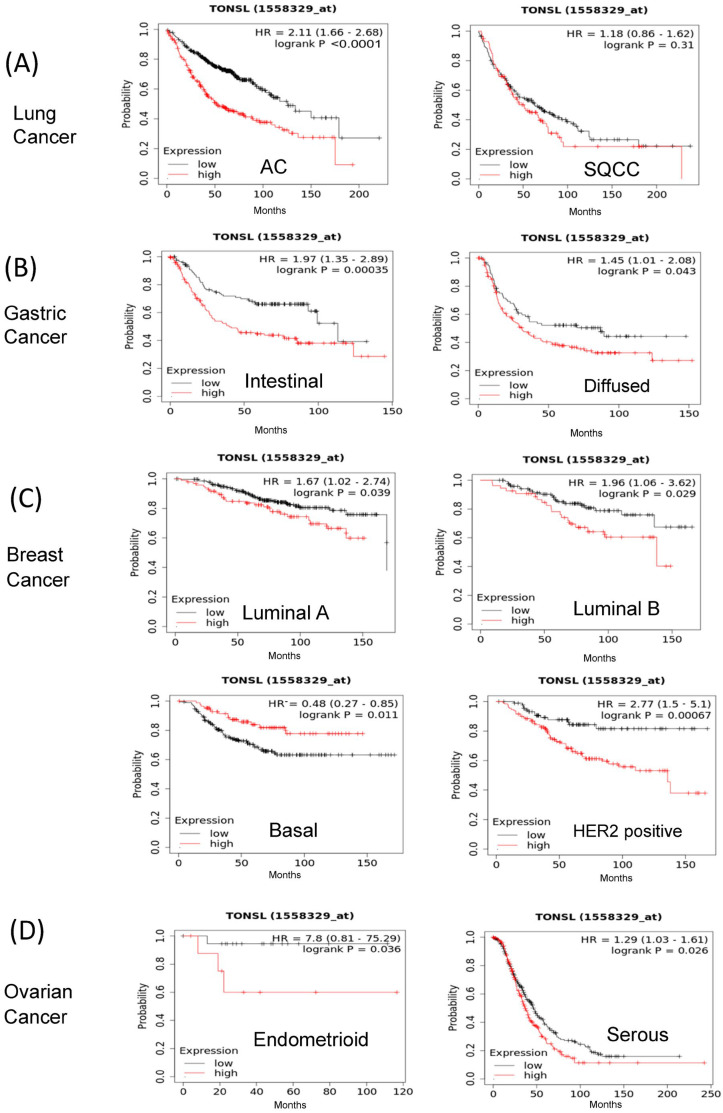
*TONSL* mRNA expression (from a gene chip) was associated with cancer prognosis (overall survival). Kaplan–Meier survival curves of *TONSL* mRNA expression levels for (**A**) lung cancer (AC, *n* = 672; SQCC, *n* = 136), (**B**) gastric cancer (intestinal, *n* = 269; diffused, *n* = 136), (**C**) breast cancer subtypes (luminal A, *n* = 377; luminal B, *n* = 177; HER2+ (*n* = 223) and basal (*n* = 278), and (**D**) ovarian cancer (endometrioid, *n* = 30; serous *n* = 523). Auto select best cutoff was used to split the cohorts of patients. All of the cohort showed a statistically significant difference. AC—adenocarcinoma; SQCC—squamous cell carcinoma; HR—hazard ratio; The log-rank test *p* value was used for statistical significance. The data in KM Plotter (https://kmplot.com/analysis/) were accessed on 1 December 2021.

**Figure 2 ijms-24-09530-f002:**
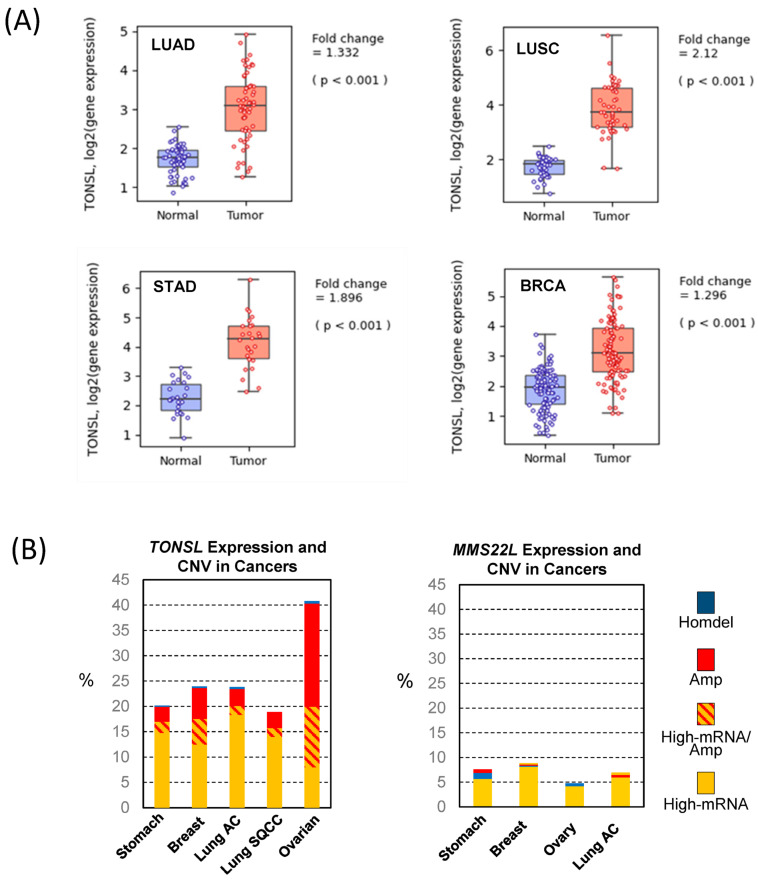
*TONSL* and *MMS22L* mRNA expression and copy number variations. (**A**) *TONSL* mRNA expression in tumors and the adjacent normal tissues. The RNA seq data were derived from the TCGA Pan-Cancer Atlas study and analyzed by Q-omics. LUAD—Lung adenocarcinoma; LUSC—Lung squamous cell carcinoma; BRCA—Breast invasive carcinoma; STAD—Stomach adenocarcinoma. The *p* value was acquired by Student’s *t*-test. (**B**) Relation between *TONSL* and *MMS22L* mRNA expression level and the copy number variations (CNV) in tumors. The analysis was performed at cBioportal using the original dataset from the TCGA Pan-Cancer Atlas Study. AC—adenocarcinoma; SQCC—squamous cell carcinoma; Homdel—Homologous deletion; Amp—Amplification.

**Figure 3 ijms-24-09530-f003:**
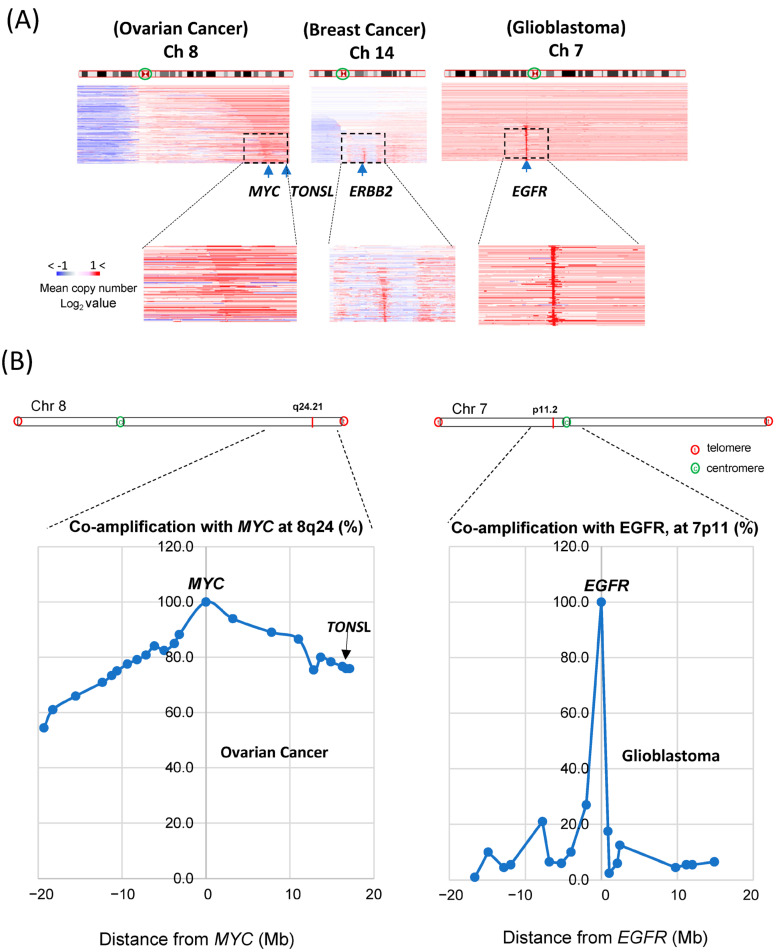
Coamplification of driver oncogenes and the adjacent genes. (**A**) Copy number variations (CNV) of three major driver oncogenes (*MYC*-Ch8, *ERBB2*-Ch14 and *EGFR*-Ch7) in tumors (Ovarian, Breast and Glioblastoma, respectively). cBioportal data (accessed on 1 December 2022) image was captured. (**B**) The percentage of coamplification of the genes close to *MYC* and *EGFR* in Ch8q24 in ovarian cancer and Ch7p11 in glioblastoma, respectively, were plotted. The coamplification was relatively high around *MYC*, and the genes close to *TONSL* were coamplified more than the genes in the opposite side. The amplification of *EGFR* is not associated with other close segment genes.

**Figure 4 ijms-24-09530-f004:**
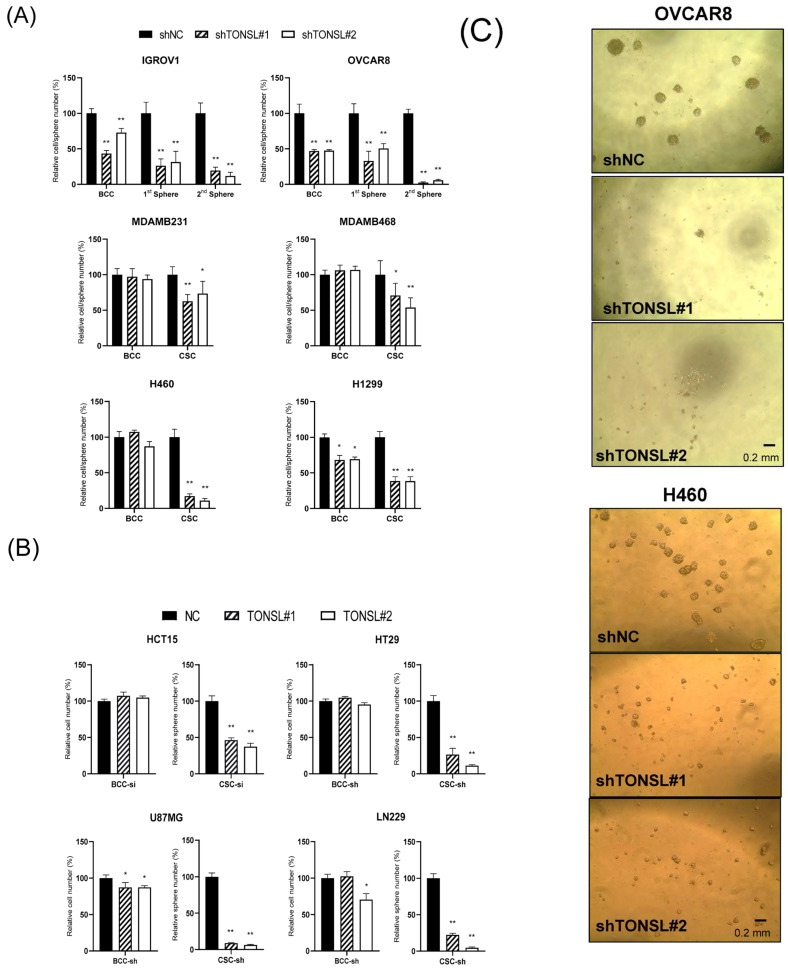
Requirement of *TONSL* expression in BCC (monolayer bulk cultured cells) and CSC (cancer stem-cell enriched cultured cells). (**A**) Cell survival/growth of BCC and sphere numbers of CSC after the knocking down of *TONSL*. Cell survival/growth was moderately or not limited by the knockdown of *TONSL* in the BCC, while it is critical in the CSC in two ovarian cancer cell lines (IGROV1 and OVCAR8), breast cancer cell lines (MDAMB231 and MDAMB468) and lung cancer cell lines (H460 and H1299). (**B**) Cell survival/growth of BCC and sphere numbers of CSC after the knocking down of *TONSL*. Cell survival/growth was moderately or not limited by the knockdown of *TONSL* in the BCC, while it is critical in the CSC in two colon cancer cell lines (HCT15 and HT29) and Glioblastomas (U87MG and LN229). (**C**) The representative images of CSC spheres. *, *p <* 0.05; **, *p <* 0.01. Student’s *t*-test vs. siNC or shNC.

**Figure 5 ijms-24-09530-f005:**
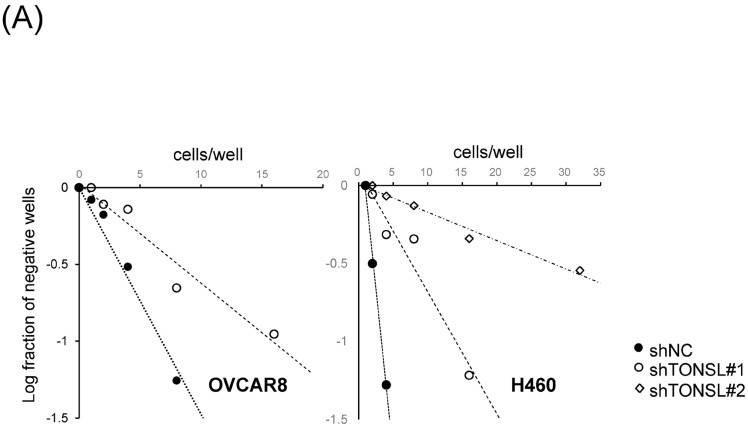
The loss of the stem cell population by TONSL depletion. (**A**) The limited dilution assay was performed using the secondary CSC spheres. The primary sphere generated by the seeding of BCC was split by accutase and reseeded into 96 wells at different cell density. After one week, the no-sphere forming wells were counted and plotted against the seeding of cells. (**B**) Aldehyde dehydrogenase (ALDH) activity assay was performed using the siRNA-treated secondary spheres of U87MG cells after 48 h of transfection on primary CSC spheres. siSCD1(26) was used as a positive control. (**C**) Cell senescence was induced by the loss of *TONSL* expression. The histogram of SA-βgal positive cells in the control shRNA and sh*TONSL*(#1)-treated OVCAR8 cells. The senescence cells increased much more in CSC than in the BCC by the sh*TONSL. ***, *p <* 0.01. Student’s *t*-test vs. siNC or shNC. DEAB is the inhibitor of ALDH activity.

**Figure 6 ijms-24-09530-f006:**
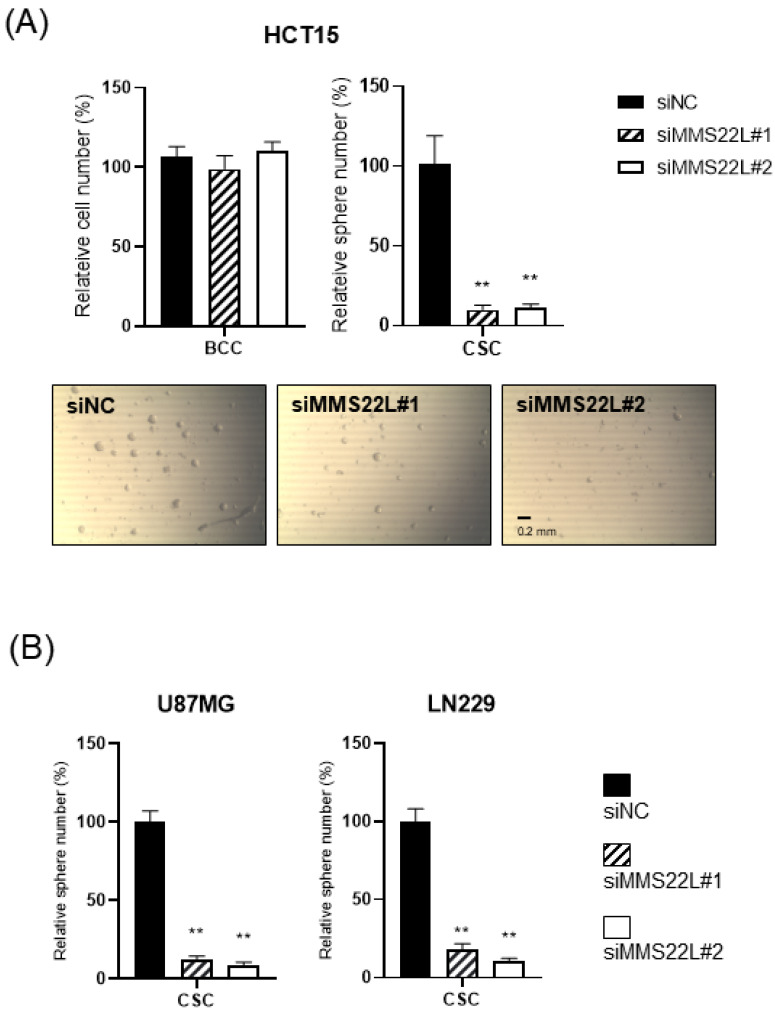
Requirement of MMS22L in colon cancer and glioblastoma CSC. (**A**) Cell survival/growth was not limited by the knockdown of MMS22L in the BCC, while they were critical in the CSC in a colon cancer cell line, HCT15. The representing images of HCT15 CSC spheres after being treated with siRNA. (**B**) CSC sphere growth requires MMS22L in glioblastomas, U87MG and LN229. ****, *p* < 0.01. Student’s *t*-test vs. siNC.

**Figure 7 ijms-24-09530-f007:**
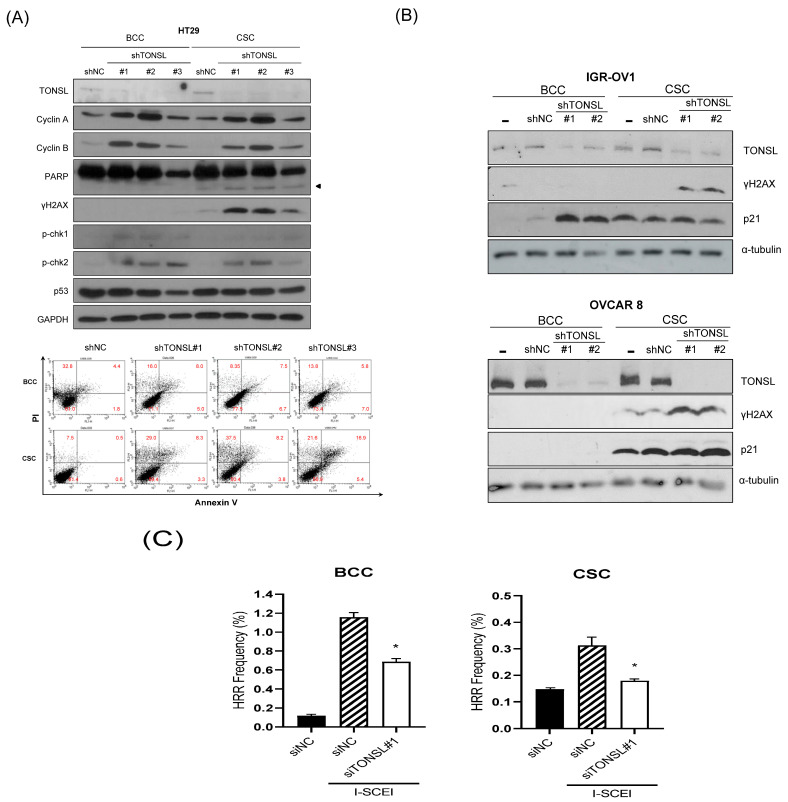
Loss of TONSL in colon cancer CSC (cancer stem cell-enriched culture) results in the DNA damage-induced apoptosis. (**A**) The biochemical pathways related to the DNA damage were activated by the sh*TONSL* infection. The damage accumulated more in CSC than in BCC, and increased the apoptosis in CSC. Represented cell apoptosis flowcytometry after shRNA of *TONSL* was infected. (**B**) gamma H2AX accumulated more in CSC by sh*TONSL* virus infection. NC—negative control sequence virus. (**C**) The homologous recombination frequency was measured by DR-GFP plasmid transfected U87MG cells after I-SCE1 endonuclease plasmid was transfected. The GFP-positive cells are counted as the HRR-positive cells. ***, *p* < 0.05 Student’s *t*-test vs. siNC.

**Figure 8 ijms-24-09530-f008:**
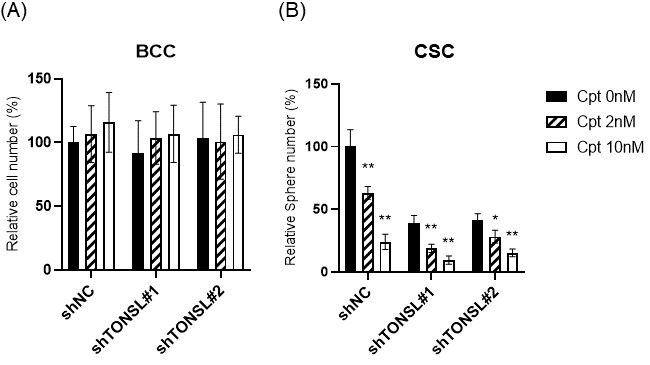
CSC requires TONSL to survive from Camptothecin (Cpt)-induced DNA damage (**A**) The BCC cell survival was not influenced by *TONSL* shRNA or Cpt (2 nM and 10 nM). (**B**) CSC requires the TONSL to recover from the toxicity by the Cpt (2 nM and 10 nM). ***, *p* < 0.05; ****, *p* < 0.01. Student’s *t*-test vs. shNC.

**Figure 9 ijms-24-09530-f009:**
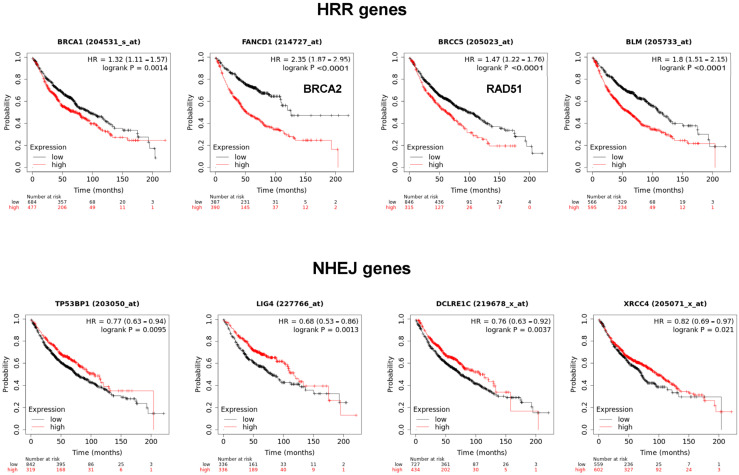
mRNA expression of homologous recombination repair (HRR) genes and nonhomologous end joining (NHEJ) genes are differentially associated with cancer prognosis of lung adenocarcinomas. Kaplan–Meier survival curves of four HRR genes (BRCA1, BRCA2, RAD51 and BLM) and four NHEJ genes (TP53BP1, LIG4, ARTEMIS and XRCC4). (The data was accessed on 25 April 2023, KM Plotter). All of the cohort showed a statistically significant difference (*p* value was obtained by the log-rank *t*-test).

## Data Availability

Any materials and data used in this research will be available on reasonable request.

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
