# Peer review of "Oncogenic Impact of TONSL, a Homologous Recombination Repair Protein at the Replication Fork, in Cancer Stem Cells"

_ijms, 2023, doi:10.3390/ijms24119530_

Round 1

Reviewer 1 Report

In this manuscript, the authors demonstrate the oncogenic role of the replication fork HRR gene TONSL, which is often amplified along with MYC, especially for the maintenance of CSCs; their findings suggest that this gene, maybe alone with MMS22L, is a potential target in CSC elimination therapy. The results also indicate that CSC-preferred HRR might be a vulnerable target in CSCs, which is reminiscent of normal stem cells. Although further investigation regarding the clinical use of these potential target processes is needed, this study definite helps the field to move to that direction. In general, this manuscript is well organized and written, I only identified some typo and stylistics issues: such as

1. Page 2, line 87; we could not fup;analyze relative expression between

2. Page 5, line157; cultur

Reviewer 2 Report

In the submitted manuscript authors studied the role of TONSL, a mediator of homologous recombination repair (HRR) in the stalled replication fork double strand breaks (DSB) in cancer using various in silico and in vitro methods.

Unfortunately, the overall quality of this manuscript is too poor so it should be rejected for numerous reasons:

1) The quality of English language is low, and lots of sentences and phrases have either wrong or unclear meaning. For example:

- promoter hypermutation

- all these cancer tissues showed more than 5% TONSL gene amplification

- tumors expressed more TONSL with gene amplification

- strong oncogenes

- powerful MYC

- homologous deletion

- the strong oncogenic driver MYC (c-myc, 8q24.2), which is located at 8q24.2

2) 'Abstract' is too abstract, and an average reader could not understand what authors actually studied, e.g., which all types of tumors were studied, which well used in vitro models, which all actual methods were used, etc.

3) This manuscript is very incoherent, and flow of thoughts is not very straightforward. For instance, it was not clearly explained why authors chose one particular set of cancer type for one bioinformatic analysis, and then second for other; while performing in vitro analyses only on ovarian, colon brain cancer cell lines?!

4) Study behind this manuscript is COMPLETELY irreproducible because many methods were either not described at all or are missing too many important details (e.g., cell treatments, qPCR conditions, primary antibodies, bioinformatical methods, statistical analyses, etc.), so readers would not understand how authors actually came with the results.

5) Some of the used cell lines are improper, e.g., according to https://www.ncbi.nlm.nih.gov/books/NBK367613/table/tab_2_2/ those two used ovarian cancer cell lines are not models of HGSOC. Also, HT29 were mentioned in 'Methods' but results on them were not presented, while it is unclear which actual problematic U-87MG cells were used (e.g., https://www.cellosaurus.org/CVCL_GP63).

6) Most figures are too small, and overcrowded with text, and thus pretty indiscernible.

7) Supplementary material is a total mess! Figures must be put in a proper size and resolution, not be overcrowded with text, proper figure legends and table captions must be provided (also explaining all presented abbreviations), Figure S4A should be presented as table, etc.

8) Authors improperly interpreted some results of statistical analyses, e.g., interpreting non-significant p-values of log-rank test (Fig.1A SQCC and Fig.9 BRCA2) Also in that light, statement "Therefore, mRNA expression of TONSL is positively associated with worse prognosis, though it may function the opposite way in some subtypes of cancer like basal type of breast cancer." is meaningless.

Reviewer 3 Report

In this manuscript, the authors investigate the role of the DNA repair protein TONSL in cancer cell proliferation/survival, focusing largely on cancer stem cells. They first examine existing clinical databases and find that high levels of TONSL mRNA expression are associated with decreased survival across several cancer types. They also find increased expression of TONSL mRNA in cancer samples compared to normal tissues regardless of gene amplification. The authors then study the effects of loss of TONSL in cancer cell lines using shRNA/siRNA knockdown. They find that knockdown of TONSL has variable effects on proliferation/survival of standard cancer cell lines, however they see a more pronounced effect when the lines are enriched for cancer stem cells. They also show that loss of TONSL is associated with increased markers of DNA damage, cell cycle arrest, and apoptosis. Finally, they return to clinical databases to show that increased expression of HR genes is associated with worse survival, whereas increased expression of NHEJ genes is associated with better survival in lung cancer.

The impetus for this study was driven by previous work from this group that identified TONSL as a candidate factor important for cancer stem cell survival. The authors attempt to expand upon this finding in the current manuscript. While the results are interesting, the study is rather superficial and would benefit from additional validation of the findings as well as mechanistic details. 

Major points

- The authors suggest that cancer stem cells are more sensitive to loss of TONSL compared to bulk cancer cell populations. However, the data they present are variable. For instance, ovarian cancer bulk and stem cell populations are both sensitive to TONSL loss across 2 cell lines, whereas in a colon cancer cell line the stem cell population is specifically sensitive. The authors should do additional experiments to address this discrepancy, such as look at additional cell lines and knockdown TONSL for longer time periods. Data from the cancer dep map (https://depmap.org/portal/gene/TONSL?tab=overview) suggest that TONSL should be essential for survival across the majority of cancer cell lines. 

- The authors should provide additional details of the methods used to look at cancer stem cells. Could the different preparations/assays used for bulk populations vs cancer stem cells could account for the differences seen in these populations? Additionally, could cell cycle differences or proliferation rate of bulk vs stem cell populations contribute to the results seen?

- The authors imply that the phenotypes seen with loss of TONSL are due to disruption of DNA repair at the replication fork. However, they only provide one western blot which is insufficient to make mechanistic conclusions. The authors should provide additional measures of DNA damage/replication problems in bulk vs stem cell populations. Possible experiments include comet assay, immunofluorescence of gH2AX/RAD51, DNA fiber assay, etc. The authors could also look at recruitment or TONSL to damage sites in the different populations.

- The authors suggest that HR is more important for cancer stem cells compared to bulk cancer cells populations. However they do not provide any evidence for this. This claim should be supported by experimental evidence such as knocking down HR vs NHEJ factors in their stem cell vs bulk populations and looking at survival/DNA damage accumulation.

Minor points

- High expression of TONSL does not correlate well with gene amplification. Is it possible that higher expression of TONSL in cancer cells is due to increased proliferation compared to normal tissue? Would the same pattern be expected for proliferating non-cancer cells?

- The expression data suggest that increased expression of TONSL in cancer cells is associated with worse survival. However, the knockdown experiments do not demonstrate a strong phenotype in bulk cancers with loss of TONSL. How do the authors reconcile this discrepancy?

- The authors should comment on any roles for TONSL outside of complex with MMS22L as their expression patterns are not the same.

- How do authors explain the finding that germline LOF mutations of HR factors contribute to cancer while increased expression of HR factors is associated with poor prognosis?

- Figure 7B does not have the appropriate labels. It’s unclear what the different plots are referring to.

- For the CPT experiment in Figure 8, how do higher doses of CPT affect viability in the BCC populations? Additionally, in the right panel the relative CPT sensitivity appears to be similar in the knockdowns compared to control. 

- A rescue of siRNA/shRNA TONSL would provide additional validation that the findings are not due to off target effects.

Round 2

Reviewer 2 Report

Authors have satisfactorily responded to all of the reviewers' concerns and substantially improved quality of this manuscript through revision.

Just few minor corrections:

- gene symbols should uniformly be written in italics

- URL for Q-omics website is not working (line 97)

- in figure 6B there are some dots behind numbers on y-axes

Reviewer 3 Report

The authors have sufficiently addressed all my comments.